# The Impact of High Temperature on Microbial Communities in Pork and Duck Skin

**DOI:** 10.3390/microorganisms11122869

**Published:** 2023-11-27

**Authors:** Dan Hai, Haisheng Jiang, Ziheng Meng, Mingwu Qiao, Tian Xu, Lianjun Song, Xianqing Huang

**Affiliations:** 1College of Food Science and Technology, Henan Agricultural University, Zhengzhou 450002, China; 15837184289@163.com (D.H.); m15890159032_1@163.com (H.J.); zihengm@163.com (Z.M.); mingwu0309@163.com (M.Q.); xutian20140626@163.com (T.X.); slj69@126.com (L.S.); 2Henan Engineering Technology Research Center of Food Processing and Circulation Safety Control, Zhengzhou 450002, China

**Keywords:** duck skin, pork skin, microbial differences, heat treatment

## Abstract

Pork skin and duck skin are highly favored by consumers in China, and high-temperature processing methods are widely employed in cooking and food preparation. However, the influence of high-temperature treatment on the microbial communities within pork skin and duck skin remains unclear. In this study, a high-temperature treatment method simulating the cooking process was utilized to treat samples of pork skin and duck skin at temperatures ranging from 60 °C to 120 °C. The findings revealed that high-temperature treatment significantly altered the microbial communities in both pork skin and duck skin. Heat exposure resulted in a decrease in microbial diversity and induced changes in the relative abundance of specific microbial groups. In pork skin, high-temperature treatment led to a reduction in bacterial diversity and a decline in the relative abundance of specific bacterial taxa. Similarly, the relative abundance of microbial communities in duck skin also decreased. Furthermore, potential pathogenic bacteria, including Gram-positive and Gram-negative bacteria, as well as aerobic, anaerobic, and facultative anaerobic bacteria, exhibited different responses to high-temperature treatment in pork skin and duck skin. These findings highlighted the substantial impact of high-temperature processing on the composition and structure of microbial communities in pork skin and duck skin, potentially influencing food safety and quality. This study contributed to an enhanced understanding of the microbial mechanisms underlying the alterations in microbial communities during high-temperature processing of pork skin and duck skin, with significant implications for ensuring food safety and developing effective cooking techniques.

## 1. Introduction

Pork skin and duck skin are commonly used food ingredients with high consumption rates and frequency in China. Currently, China ranks first in terms of pig population, slaughter rate, and pork production, with an average annual per capita pork consumption of 40 kg [1]. Pork skin, which constitutes approximately 10% of the total weight of a pig carcass, is a valuable by-product of pork production. It is nutritionally rich, particularly in collagen protein, which can be converted into gelatin with a mesh-like structure after cooking, providing physiological benefits to the human body [2,3]. Pork skin has broad market prospects and application value. It can be added to sausages, hams [4], and other meat products to enhance their texture, or processed into specialty products such as pickled pork skin and fried pork skin [5]. Moreover, delicious dishes made primarily from pork skin, such as braised pork skin, pork skin jelly, and pork skin and red date soup, are highly favored by people [6]. Pork skin, which contains 2.5 times the protein content of pork and is rich in fat, is highly susceptible to microbial contamination during storage and transportation [7]. Moreover, the processing of pork skin involves manual operations such as peeling and segmentation, further increasing the risk of microbial contamination [8]. Coupled with the nutritional richness of pork skin, a diverse range of bacteria can easily form on its surface. When pork skin is contaminated by bacteria, under certain temperature conditions, bacteria on the surface of pork skin can rapidly multiply and proliferate, leading to poor sensory quality, spoilage, and a significant impact on the shelf life of pork skin and its products, resulting in substantial economic losses [9,10].

Duck skin is a by-product of duck slaughtering processes, valued for its nutritional content and its capacity to enhance properties such as color, texture, and flavor in meat products [11]. In contrast to developed countries, China exhibits relatively limited utilization of poultry by-products, with the extent and depth of utilization varying among enterprises of differing scales [12,13,14,15]. Historically, the primary focus for utilizing pork skin, duck skin, and chicken skin has been the extraction of collagen proteins [16]. However, recent years have witnessed a substantial shift, with food companies and researchers emphasizing a more comprehensive approach to the development and utilization of animal skins to meet the demands of an increasingly discerning consumer base. This includes innovations like the creation of novel and diverse ham sausages [17,18].

In China, the poultry slaughtering and processing process involves a series of steps, including hanging poultry, stunning with electric shock, slaughter, bleeding, scalding, depilation, body surface inspection, gizzard removal, feather cleaning, decapitation, evisceration, visceral inspection [19,20], separation of carcass by-products, by-product processing, pre-cooling inspection, high-pressure rinsing of carcasses, pre-cooling and decontamination, post-precooling inspection, and segmentation [21]. The pre-cooling and decontamination stage constitutes a critical control point for microbial management during poultry meat processing. Effective pre-cooling can regulate the growth of microbial organisms in duck meat and related products. The microbial load of duck skin raw materials during the pre-cooling stage is closely linked to the initial bacterial count during the thermal processing stage, and the reduction in both the variety and quantity of microorganisms in duck skin significantly influences the control of meat product safety [22].

Currently, heat treatment remains a common method for sterilization in the food industry. Depending on the temperature, meat products can be categorized as low, medium, or high-temperature processed, with a sterilization temperature range of 60 to 120 °C. Different temperatures yield varying sterilization effects [23]. Different types of bacteria exhibit varying degrees of heat tolerance, which can lead to variations in the spoilage bacterial communities in meat products subjected to different heat treatments. High-throughput sequencing technologies include a step involving polymerase chain reaction (PCR), which cannot distinguish between DNA from live and dead cells, potentially resulting in unrealistically high diversity estimates or overestimation of the number of live cells. However, propidium monoazide (PMA) contains azido groups that, when exposed to light, can covalently cross-link with DNA from dead cells, thus inhibiting the amplification of DNA from dead cells. PMA is incapable of entering live cells and covalently cross-links only with DNA from dead cells. Leveraging this advantage, PMA molecules are used to treat the DNA from dead bacteria in duck skin and pork skin after heat sterilization [24,25,26], thereby obtaining the composition of the bacterial communities that remain alive in duck skin and pork skin.

This approach is of significant importance in guiding the selection of sterilization processes for heat-treated meat products, as well as in the prevention and prediction of potential spoilage microorganisms.

## 2. Materials and Methods

### 2.1. Materials and Reagents

Fresh pork skins were sourced from a company located in Henan, China. The pork skins were collected following a standardized procedure. Similarly, fresh duck skin was obtained from a meat product company in Anyang, Henan Province, China, specifically using Beijing ducks. Both samples were handled and processed in accordance with the guidelines provided by the respective companies. NaCl, a key reagent used in the experiments, was purchased from Jiangtian Chemical Technology Co., Ltd., Tianjin, China. The DNA extraction kit was purchased from Beijing Tian En Ze Gene Technology Co., Ltd., Beijing, China. TransStart FastPfu DNA Polymerase was acquired from Beijing Tsingke Biotech Co., Ltd., Beijing, China. The 100 to 2000 bp DNA marker was obtained from Shanghai Sangon Biotech Co., Ltd., Shanghai, China. All other reagents and chemicals were purchased from Henan Bigen Biotechnology Co., Ltd., Zhengzhou, China and were of analytical grade. EX Taq enzyme was purchased from TaKaRa, Kusatsu, Shiga, Japan. Propidium Monoazide (PMA) was obtained from Biotium, Fremont, CA, USA. Dimethyl sulfoxide (DMSO) was purchased from Bigen Biotechnology Co., Ltd., Zhengzhou, China.

### 2.2. Instruments and Equipment

CFX 96 Real-Time PCR Detection System was from Bio-Rad, Hercules, CA, USA. FXB303-2 Constant Temperature Incubator was from Shanghai Shuli Instrument Co., Ltd., Shanghai, China. NanoDrop 2000 Spectrophotometer and 1300 Series Class II Type A2 Biological Safety Cabinet were from Thermo Fisher Scientific (Suzhou) Co., Ltd., Suzhou, China. Eppendorf 5430 High-Speed Centrifuge was from Eppendorf, Hamburg, Germany. ABI 9700 PCR System was from GeneAmp, USA. DYY-6C Electrophoresis Apparatus was from Beijing Liuyi Instrument Factory, Beijing, China. Gel DocIt TS3 UV Gel Imaging System was from UVP, USA. SX-500 Fully Automatic Steam Sterilizer was from Tomy, Tokyo, Japan. BagMixer 400CC Homogenizer was from BagMixer, Issy-les-Moulineaux, France. Qubit 2.0 Fluorometer was from Life Technologies, Shanghai, China. Agilent 2100 Bioanalyzer was from Agilent Technologies, Santa Clara, CA, USA. MiSeq System was from Illumina, San Diego, CA, USA.

### 2.3. Experimental Methods

#### 2.3.1. Sample Preparation and Grouping

Under sterile conditions, the pork skin and duck skin samples were divided into 24 random portions, each weighing 25 g. These portions were then placed in sterile homogenization bags and securely sealed with rubber bands. The sterile homogenization bags, each containing the pork skin and duck skin samples, were further divided into 3 groups, with 8 portions in each group. Each group was subjected to water bath heat sterilization at temperatures of 25, 60, 70, 80, 90, 100, 110, and 120 °C, following a standardized experimental procedure.

A needle-type meat center thermometer (range: 50–300 °C) was directly inserted into the duck and pig skin. When the center temperature of the sample reached the desired target temperature, a 10 min timer was initiated. Subsequent to the heat treatment, the samples were placed in a constant temperature incubator at 25 °C for 1 week to facilitate the enrichment of viable bacteria.

Following the 1-week incubation period, the samples were retrieved, and 225 mL of sterile saline solution was added to each sample. The samples were homogenized for 120 s using a stomacher. A 20 mL aliquot of the homogenized mixture was obtained by centrifugation at 17,226× *g* for 10 min to collect the bacteria. The supernatant was discarded, and the collected bacteria were resuspended in 500 μL of sterile saline solution (0.9% NaCl) and set aside.

#### 2.3.2. PMA Treatment

To prepare the PMA (Propidium Monoazide) treatment, 1 mg of PMA was dissolved in 200 μL of 20% (*v*/*v*) dimethyl sulfoxide (DMSO) to create a stock solution of 5 μg/μL, which was stored at −20 °C in a light-protected environment. For experimental use, the PMA stock solution was diluted tenfold to generate a working PMA solution.

Using a transparent 1.5 mL centrifuge tube, 10 μL of the PMA working solution was added to 500 μL of the bacterial solution, resulting in a final mass concentration of 10 μg/mL. The mixture was then incubated in the dark at room temperature for 5 min. Subsequently, the sample was exposed to a 650 W halogen light source for 5 min, positioned approximately 20 cm away from the sample tube, which was placed horizontally on ice to prevent overheating. The sample tube was gently shaken during this process to ensure uniform illumination. After light-induced cross-linking, the sample was centrifuged at 17,226× *g* for 10 min to collect the bacteria.

#### 2.3.3. Bacterial Genomic DNA Extraction

In total, 225 mL of sterile physiological saline solution was added to the sterile homogenization bags containing the duck skin and pork skin samples. The mixture was homogenized for 120 s using a stomacher. After homogenization, a 20 mL aliquot of the homogenized mixture was collected by centrifuging at 12,000 rpm for 10 min. Subsequently, bacterial genomic DNA was extracted from the precipitate using a bacterial DNA extraction kit.

#### 2.3.4. Amplification of 16S rDNA Gene

Genomic DNA was amplified using the polymerase chain reaction (PCR) with TransGen AP221-02: TransStart Fastpfu DNA Polymerase, in a 20 μL reaction system. The components used were 5 × FastPfu Buffer (4 μL), 2.5 mmol dNTPs (2 μL), Forward Primer (5 μmol) (0.8 μL), Reverse Primer (5 μmol) (0.8 μL), FastPfu Polymerase (0.4 μL), BSA (0.2 μL), Template DNA (10 ng), and supplemented with ddH_2_O to make a final volume of 20 μL. The PCR reaction parameters were as follows: initial denaturation at 95 °C for 3 min, followed by 27 cycles consisting of denaturation at 95 °C for 30 s, annealing at 55 °C for 30 s, extension at 72 °C for 45 s, and a final extension at 72 °C for 10 min. The reaction was then held at 10 °C until further use.

#### 2.3.5. Sequence Bioinformatics Analysis

The amplified DNA was sent to Shanghai Meiji Biomedical Technology Co., Ltd., Shanghai, China for sequencing using the Illumina MiSeq platform. The obtained sequences were subjected to bioinformatics analysis using FLASH and Trimmomatic software (Version 0.38).

In the QIIME platform (http://qiime.org/scripts/assign_taxonomy.html, accessed on 6 June 2022), the Bayesian algorithm of the RDP classifier (Version 2.2, http://sourceforge.net/projects/rdp-classifier/, accessed on 12 August 2022) was employed for taxonomic analysis of representative sequences of OTUs at the 97% similarity level [27,28,29]. Taxonomic composition of each sample was determined at the phylum and genus levels and compared with the Greengene bacterial 16S rRNA database (Release 13.5, http://greengenes.secondgenome.com, accessed on 23 October 2022) [30].

## 3. Results and Discussion

### 3.1. Analysis of Relative Abundance of Bacterial Genera at the Phylum Level in Duck Skin and Pork Skin

Figure 1 illustrates the microbial composition in the control groups (25 °C) of pork skin and duck skin. In pork skin, the dominant genus was *Psychrobacter*, constituting 39.8% of the microbiota, whereas in duck skin, *Pseudomonas* was the dominant genus, comprising 30.65% of the total microbial population.

In the high-temperature treatment of pork skin ranging from 60 °C to 120 °C, the predominant genus shifted significantly. At 60 °C, *Bacillus* emerged as a major genus, accounting for 51.7% of the relative abundance, accompanied by a lower presence of *Pseudomonas* at 8.32%. This dominance by *Bacillus* escalated at 70 °C, reaching an impressive 98.9% relative abundance. At 80 °C and 90 °C, *Bacillus* remained the dominant genus, with average relative abundances of 85.9% and 85.0%, respectively. *LysiniBacillus* emerged as the subdominant genus, with relative abundances of 26.3% and 30.9% at 80 °C and 90 °C, respectively. The influence of *Bacillus* waned at 100 °C, accounting for 8.3% of the relative abundance. However, at 110 °C and 120 °C, *Bacillus* regained dominance with relative abundances of 72.5% and 86.4%, respectively.

At lower temperatures, such as 60 °C, *Bacillus* species were present but not as dominant, allowing other genera, like *Pseudomonas*, to coexist. However, as the temperature escalated to 70 °C and beyond, *Bacillus*’s competitive advantage became increasingly evident. The spore-forming ability of *Bacillus* species, which enables them to withstand harsh conditions, including high temperatures, likely contributed to their dominance in these environments.

The temporary decrease in *Bacillus* abundance at 100 °C may be attributed to the extreme conditions at this temperature, potentially affecting the spore formation and survival of certain *Bacillus* species. However, at 110 °C and 120 °C, *Bacillus* species rebounded, showcasing their adaptability and resilience and enabling them to reestablish dominance within the microbial community.

In the case of duck skin subjected to temperatures from 60 °C to 80 °C, the microbial composition exhibited a more diversified distribution without a single dominant genus. Several genera contributed to the composition. At 60 °C, the major genera included *Pseudomonas* (33.52%) and *Acinetobacter* (18.55%), along with lower abundances of *Serratia* (5.71%) and *Clostridium sensu stricto 7* (6.32%). At 70 °C, the dominant genera comprised *Acinetobacter* (18.46%), *Clostridium sensu stricto 18* (12.41%), *Serratia* (11.04%), *Pseudomonas* (10.17%), and *Clostridium sensu stricto 7* (8.62%). The 80 °C and 90 °C treatment groups were characterized by the dominance of *Clostridium sensu stricto* 18, with relative abundances of 63.04% and 84.11%, respectively. At 100 °C, the dominant genera included *Bacillus* (49.50%), *Clostridium sensu stricto* 18 (29.19%), and *Anaerosalibacter* (17.20%). Subsequently, at 110 °C, *Bacillus* emerged as the dominant genus, constituting 99.23% of the microbial population, while at 120 °C, the dominant genera comprised Clostridium sensu stricto 18 (52.41%), *Bacillus* (33.07%), and *Clostridium sensu stricto 7* (12.23%). The contrast in *Bacillus* dominance between 110 °C and 120 °C highlights the remarkable adaptability of this genus to extreme thermal conditions.

At 110 °C, *Bacillus* demonstrates its exceptional resilience and becomes the overwhelmingly dominant genus, comprising nearly the entire microbial community. This dominance can be attributed to *Bacillus*’s ability to form spores and endure extreme temperatures. In contrast, at 120 °C, the microbial community undergoes a shift, with *Bacillus* still present but sharing dominance with other genera. The co-dominance of *Bacillus* and Clostridium sensu stricto 18 and the emergence of *Clostridium sensu stricto 7* at 120 °C suggest a more complex microbial response to the extreme conditions.

### 3.2. Analysis of Differences between Gram-Positive and Gram-Negative Bacteria

Gram-positive (Figure 2a) and Gram-negative (Figure 2b) column charts were employed to analyze and compare the relative abundance of various genera present in duck skin and pork skin samples. These visualizations facilitate the assessment of abundance variations among different genera and their distribution across diverse sample sets.

Gram-Positive Bacteria (Figure 2a):

Examining Figure 2a, it is evident that the dominant genus in both duck skin and pork skin samples within the control group (25 °C) is Vagococcus, with relative abundances of 17.07% and 24.89%, respectively. In duck skin, the relative abundance of Gram-positive bacteria exhibits an upward trend as the temperature rises. Among the 60 °C heat-treated duck skin samples, Gram-positive bacterial distribution is relatively uniform, with *Clostridium sensu stricto 7* (8.99%) emerging as the dominant genus, followed by Vagococcus (6.27%). As the temperature continues to increase, the relative abundance of Gram-positive bacteria in duck skin experiences a continuous escalation. The genus *Bacillus* demonstrates an initial surge followed by a decrease in relative abundance, with levels of 30.81%, 53.73%, 88.48%, and 69.45% at 90 °C, 100 °C, 110 °C, and 120 °C, respectively. In contrast, the relative abundance of Gram-positive bacteria in pork skin gradually decreases with the elevation of temperature. The genus *Bacillus* exhibits an ascending pattern from 60 °C to 80 °C, reaching its zenith at 78.47% at 80 °C. Conversely, in heat-treated pork skin samples ranging from 90 °C to 120 °C, the abundance of *Bacillus* initiates a descent, reaching its nadir of 6.27% at 120 °C.

Gram-Negative Bacteria (Figure 2b):

Figure 2b underscores that the dominant Gram-negative genera within the control group of duck skin include *Citrobacter* (20.63%) and *Pseudomonas* (13.01%). On the other hand, the control group of pork skin showcases prevailing genera *Pseudomonas* (20.73%) and *Psychrobacter* (14.23%). Upon evaluating the 60 °C heat-treated duck skin samples, the dominant genera are *Citrobacter* (17.91%) and Enlydrobacter (16.63%). With an increase in temperature, both the relative abundance of *Citrobacter* and Gram-negative bacteria in duck skin display a gradual decline. At 70 °C and 80 °C, the relative abundance of *Citrobacter* registers as 17.32% and 9.22%, respectively. Upon reaching 90 °C and 100 °C, the relative abundance of Gram-negative bacteria experiences a continued decrease, ultimately resulting in the absence of *Citrobacter*. The trend persists at 110 °C and 120 °C, where the relative abundance of Gram-negative bacteria also reaches negligible levels. Notably, in pork skin samples, the relative abundance of Gram-negative bacteria is non-existent in the heat-treated range of 60 °C to 90 °C. At 100 °C, the relative abundance of *Psychrobacter* accounts for 31.83%. The figure subsequently declines to 10.43% at 110 °C; however, at 120 °C, the relative abundance of *Psychrobacter* suddenly surges to 53.42%.

This comprehensive analysis utilizing Gram-positive and Gram-negative bacterial classifications provides valuable insights into the behavior of different bacterial genera in duck and pork skin samples under varying thermal conditions. The results underscore the temperature-sensitive dynamics of bacterial populations, highlighting their adaptation and response to heat treatments. These findings contribute to a deeper understanding of bacterial distribution and serve as a foundation for optimizing food safety strategies in both duck and pork processing.

Based on the analysis of Figure 2, it can be concluded that the relative abundance of Gram-negative bacteria is 0 in duck skin at 110 °C heat treatment and in pork skin from 60 °C to 90 °C heat treatment. The relative abundance of Gram-negative bacteria in duck skin decreases as the temperature increases, while in pork skin, the relative abundance of Gram-negative bacteria initially increases and then decreases, reaching the highest level at 120 °C. This indicates differences in the presence of Gram-negative bacteria between duck skin and pork skin samples.

Regarding Gram-positive genera, the dominant genera in the control groups of both duck skin and Pork skin are Vagococcus, while in the treated groups, the dominant genera are *Bacillus* for both duck skin and pork skin. The relative abundance of Gram-positive bacteria in duck skin samples increases continuously as the temperature rises, with the highest abundance of *Bacillus* at 110 °C. On the other hand, the relative abundance of Gram-positive bacteria in pork skin gradually decreases with increasing temperature, with the highest abundance of *Bacillus* at 80 °C. In the examination of Gram-negative genera, the prevailing genus identified in duck skin samples is *Citrobacter*, whereas in pork skin samples, a different trend emerges. Proper temperature control during food processing and cooking is of paramount importance for the survival and distribution of microorganisms. Varied temperature treatments can lead to shifts in the relative abundance of different bacterial types within food samples, consequently influencing the quality and safety of food products. The formulation of precise food processing strategies is imperative to ensure the integrity and safety of food items.

### 3.3. Analysis of Differences in Aerobic, Anaerobic, and Facultative Anaerobic Bacteria

Aerobic Bacteria (Figure 3a):

The comparison of aerobic bacteria in duck skin and pork skin samples at different storage temperatures reveals distinctive trends. In duck skin, the abundance of aerobic bacteria progressively diminishes with rising temperatures, eventually reaching negligible levels at 110 °C and 120 °C. Conversely, pork skin samples exhibit a nuanced pattern, where the initial reduction in aerobic bacteria is succeeded by an increase. This shift coincides with the transition from the Micrococcus genus to *Psychrobacter* as the dominant genus.

In the control group of duck skin, *Pseudomonas* dominates with the highest relative abundance (13.01%), followed by Alcaligenes (11.27%). Throughout the heat treatment process, Enhydrobacter abundance steadily declines between 60 °C and 90 °C, displaying slight recovery at 100 °C before dropping to undetectable levels at 110 °C and 120 °C.

Conversely, the control group of pork skin displays higher relative abundances of *Pseudomonas* (20.74%) and *Psychrobacter* (14.23%). Following heat treatment, pork skin samples are characterized by Micrococcus dominance at 60 °C (20.24%) and 70 °C (11.73%). A decline occurs at 80 °C, with a subsequent increase at 90 °C. Notably, the abundance of aerobic bacteria initially decreases, followed by a subsequent increase, peaking at 120 °C (53.42%). This increase corresponds with the shift from Micrococcus to *Psychrobacter* as the dominant genus.

Anaerobic Bacteria (Figure 3b):

The comparison of anaerobic bacteria in duck skin and pork skin samples exposes variations. Overall, pork skin demonstrates lower anaerobic bacterial abundance. In duck skin, the abundance of anaerobic bacteria experiences an initial rise, followed by a decline, with peak levels recorded at 80 °C.

In the control group of duck skin, *Peptoniphilus* dominates with a relative abundance of 4.31%. Notable transformations occur in the anaerobic bacteria composition during heat treatment. At 60 °C, *Clostridium sensu stricto 7* prevails (8.99%), followed by a shift to Lachnoclostridium (19.92%) at 70 °C. At 80 °C, anaerobic bacteria distribution evens out, with Hathewaya exhibiting higher abundance (13.74%). The dominant phylum shifts to Anaerosalibacter at 90 °C and 100 °C, with relative abundances of 31.09% and 24.68%, respectively. At 120 °C, *Clostridium sensu stricto 7* resurfaces as the dominant genus with a relative abundance of 26.25%.

Contrarily, the control group of pork skin demonstrates lower anaerobic bacteria abundance, with no prevailing genus. Under distinct heat treatment conditions, anaerobic bacteria levels remain consistently low. An initial decrease is succeeded by a sharp increase, followed by subsequent decreases. At 90 °C, Clostridium sensu stricto 13 experiences a sudden surge, becoming the dominant genus (11.88%). Anaerobic bacteria abundance reaches zero at 100 °C, slightly recovers at 110 °C, and maintains a minor resurgence at 120 °C.

Facultative Anaerobic Bacteria (Figure 3c):

The abundance of facultative anaerobic bacteria in duck skin decreases progressively with increasing temperature, ultimately reaching absence at 110 °C. In pork skin, facultative anaerobic bacteria initially increase and peak at 80 °C (65.97%), subsequently decreasing. The decrease persists at 110 °C and 120 °C, maintaining levels lower than those in the control group.

In the control group of duck skin, *Citrobacter* dominates with a relative abundance of 20.63%, followed by Vagococcus at 11.33%. With rising temperatures, *Citrobacter* abundance gradually decreases at 60 °C, 70 °C, and 80 °C (17.91%, 17.32%, 9.22%, respectively). At 90 °C, the dominant genus shifts to *Bacillus*. As temperatures reach 100 °C and 110 °C, *Bacillus* abundance decreases to 1.00% and 0%, respectively. At the highest temperature of 120 °C, *Bacillus* abundance slightly recovers to 0.17%. Conversely, *Citrobacter* abundance in duck skin decreases with increasing temperature, yielding *Bacillus* as the dominant genus with continuous decline.

For the control group of pork skin, Vagococcus prevails with a relative abundance of 17.62%, followed by Leucobacter at 6.20%. As temperatures rise, *Bacillus* becomes the dominant genus, peaking at 65.97% at 80 °C. However, at 90 °C and 100 °C in pork skin samples, *Bacillus* abundance diminishes to 63.90% and 42.36%, respectively. In pork skin samples at 110 °C and 120 °C, *Bacillus* abundance continues to decline to 13.43% and 5.71%, respectively.

### 3.4. Identification and Analysis of Potential Pathogenic Bacteria

In recent times, concerns regarding food safety due to potential pathogenic bacteria have heightened public awareness. This study investigates the relative abundance of potential pathogenic bacteria in duck skin and pork skin samples across a spectrum of temperatures (Figure 4). Notably, this analysis sheds light on the dynamic behavior of these bacteria under varying thermal conditions.

Potential Pathogenic Bacteria in Control Groups:

In the control groups, dominant potential pathogenic bacteria in duck skin were classified under the genera *Citrobacter* (20.63%), Vagococcus (11.33%), and *Pseudomonas* (13.01%). Conversely, potential pathogenic bacteria prevalent in pork skin included *Psychrobacter* (14.23%), Vagococcus (17.62%), and *Pseudomonas* (20.74%).

Heat Treatment Effects:

Throughout the heat treatment process, the once-dominant genus *Citrobacter* among potential pathogenic bacteria in duck skin exhibited a consistent decline from 60 °C to 80 °C, culminating in absence at 120 °C. This absence was replaced by *Clostridium sensu stricto 7* (26.25%). For pork skin, *Bacillus* emerged as the dominant genus among potential pathogenic bacteria across temperatures from 60 °C to 90 °C, constituting 53.11%, 69.69%, 76.92%, and 69.63%, respectively. A shift occurred at 100 °C, where *Bacillus* and *Psychrobacter* displayed analogous abundance levels at 31.65% and 31.83%, respectively. The shift continued at 120 °C, with *Psychrobacter* becoming the dominant genus (53.42%) while *Bacillus* declined.

An intricate examination of Figure 5 uncovers potential pathogenic bacteria in duck skin, encompassing *Pseudomonas*, *Proteus*, *Acinetobacter*, *Brochothrix*, *Serratia*, *Clostridium sensu stricto 5*, *Erysipelothrix*, *Myroides*, *Escherichia-Shigella*, *Providencia*, *Bacteroides*, *Citrobacter*, and *Peptoniphilus*. For pork skin, potential pathogenic bacteria comprise *Pseudomonas*, *Psychrobacter*, *Acinetobacter*, *Proteus*, *Serratia*, *Brochothrix*, *Staphylococcus*, *Clostridium sensu stricto 5*, *Peptoniphilus*, and *Enterococcus*. While these bacteria share the classification of potential pathogens owing to their capability to induce diseases, their specific pathogenic traits and conditions can exhibit variability due to multifarious factors. Thorough research and analysis are essential to categorize them as opportunistic or foodborne pathogens.

Temperature-Dependent Abundance Changes:

A significant finding pertains to the shifts in relative abundance of potential pathogenic bacterial genera in both duck skin and pork skin as temperature fluctuates. Remarkably, potential pathogenic bacteria in pork skin maintain relatively elevated abundance levels, contrary to the trend seen in duck skin where their relative abundance diminishes with escalating temperature. Among them, the abundance of potential pathogenic bacteria in pork skin remained at a relatively high level, while the relative abundance of potential pathogenic bacteria in duck skin decreased as the temperature increased. Especially in the high-temperature treatment group, the relative abundance of *Bacillus* in pork skin decreased and was replaced by *Psychrobacter*. This suggests that although high-temperature treatment has a certain inhibitory effect on the survival and proliferation of *Bacillus*, preventive measures against *Psychrobacter* should be formulated during the processing and production of pork skin. This investigation highlights the nuanced behavior of potential pathogenic bacteria within duck and pork skin samples across diverse temperature ranges. The findings shed light on the intricate interactions between microbial community structures and their respective niches within the pork and duck skin environments, influenced by the environmental factor of temperature. These findings underscore the imperative need for the development of tailored food safety strategies and production protocols. The temperature-dependent dynamics of potential pathogenic bacteria emphasize the importance of understanding and managing microbial ecology in food processing and safety protocols.

## 4. Conclusions

This study elucidates the alterations in the microbial composition in response to different temperature conditions in duck and pork skin. Notably, high-temperature treatment of pork skin results in a prevalence of *Bacillus* species, while duck skin demonstrates a more evenly distributed bacterial population. Furthermore, the influence of temperature is evident in the relative abundance of Gram-positive and Gram-negative bacteria, as well as aerobic and anaerobic bacteria. Moreover, it is evident that temperature has a significant impact on the presence of potentially pathogenic bacteria, underscoring the critical role of temperature control in ensuring food safety. Subsequent research endeavors should aim to delve deeper into the underlying mechanisms governing temperature-induced shifts in microbial dynamics and explore the roles of microbial consortia within skin environments.

## Figures and Tables

**Figure 1 microorganisms-11-02869-f001:**
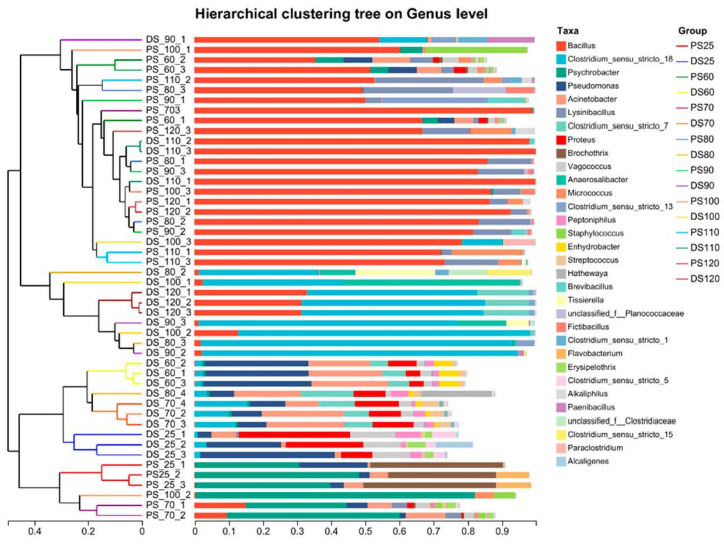
Displays the analysis of the relative abundance of microbial genera at the species level in duck skin (DS) and pork skin (PS).

**Figure 2 microorganisms-11-02869-f002:**
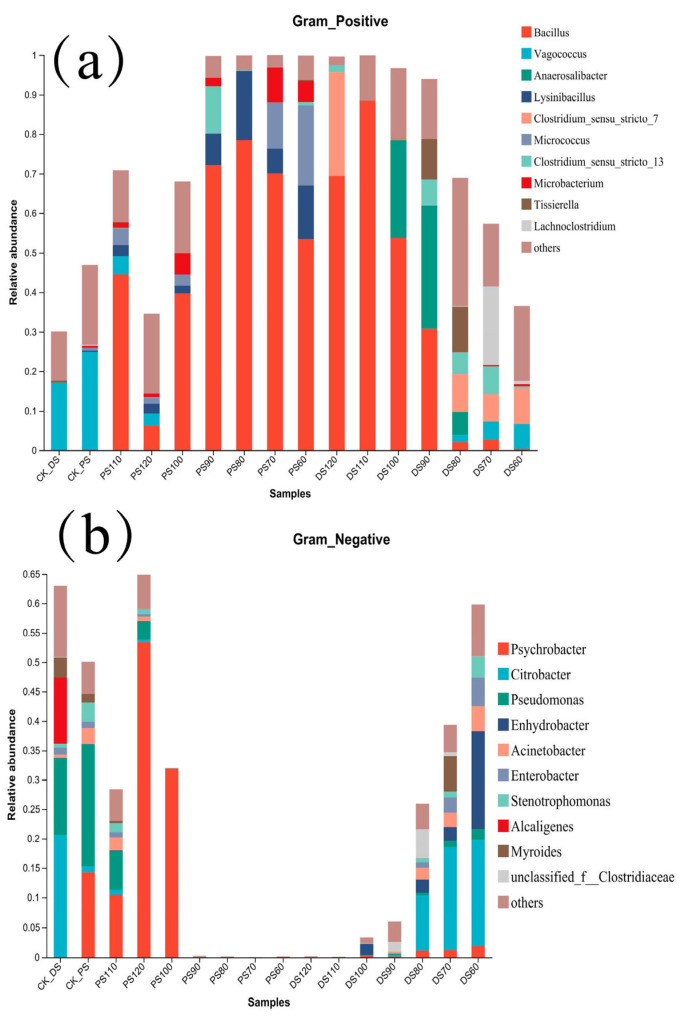
Presents the analysis of Gram-positive (**a**) and Gram-negative (**b**) microbial communities.

**Figure 3 microorganisms-11-02869-f003:**
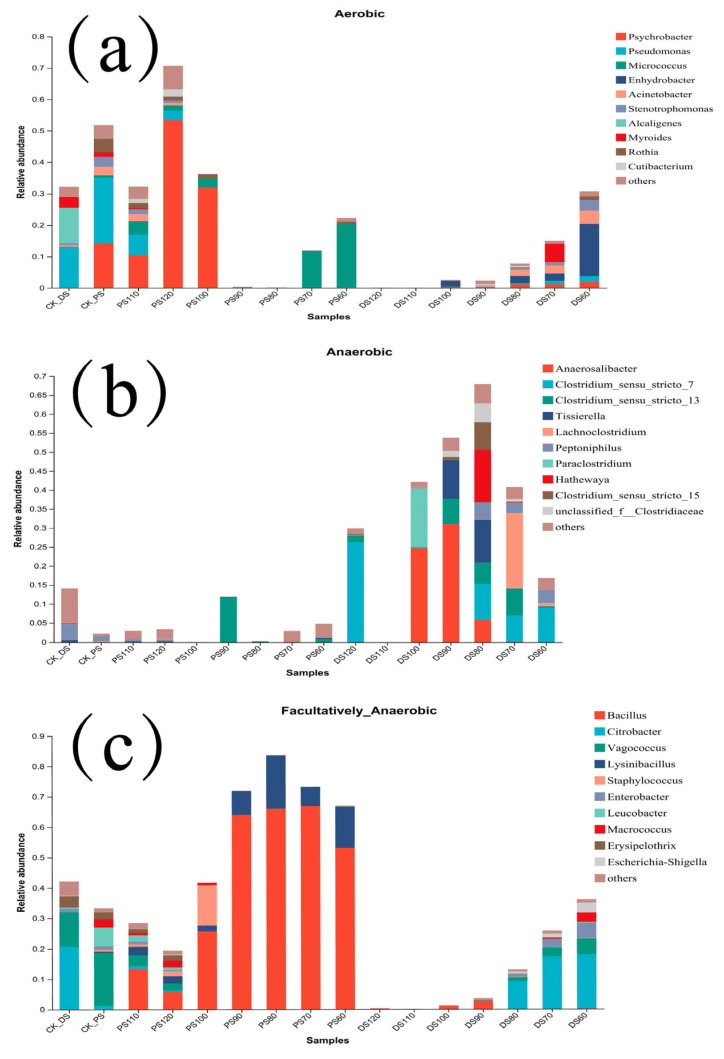
Comparison of aerobic (**a**), anaerobic (**b**), and facultative anaerobic (**c**) bacterial quantities in the microbial composition of duck skin and pork skin.

**Figure 4 microorganisms-11-02869-f004:**
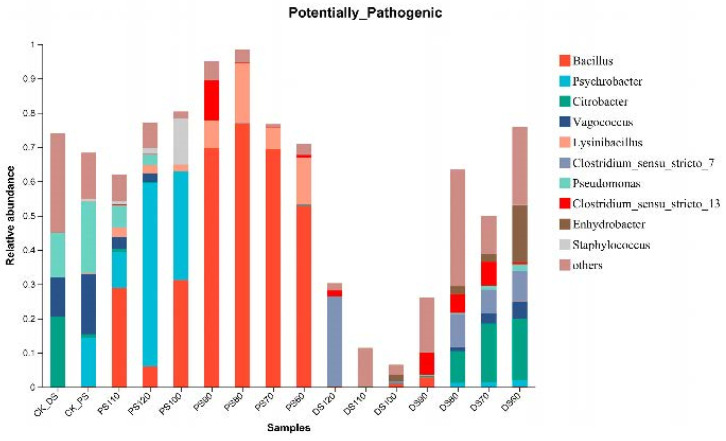
Analysis of the relative abundance of potential pathogenic bacteria.

**Figure 5 microorganisms-11-02869-f005:**
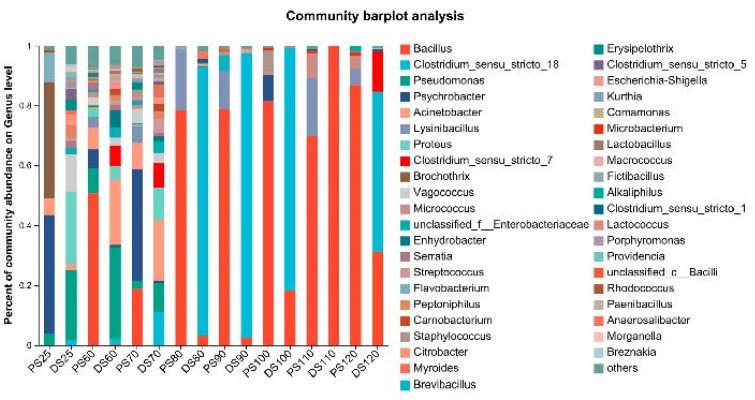
Relative abundance changes of surface microbiota based on the genus level.

## Data Availability

The data presented in this manuscript are available upon reasonable request. Interested researchers may contact the corresponding author (Dan Hai) for access to the data and any additional information regarding data availability.

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
