# Peer review of "The Impact of High Temperature on Microbial Communities in Pork and Duck Skin"

_microorganisms, 2023, doi:10.3390/microorganisms11122869_

Round 1
Reviewer 1 Report (New Reviewer)
Comments and Suggestions for Authors
Cooking is widely recognized as the primary method for ensuring the quality and microbiological safety of food. It reduces the microbial load and can even alter the microbiota profile. Since this is a well-established concept, I believe there's nothing new about it in the study. The only interesting thing is to find out which microbial genera were impacted by or resisted the treatment.
Some suggestions to further enhance the quality of the manuscript:
Title
Replace “pork skin and duck skin” by “pork and duck skin”.
Introduction
Second paragraph: it is too long, so it could be summarized or divided into two parts.
Third paragraph (last line): which microorganisms? The phrase refers to something that has not yet been presented.
Materials and Methods
Item 2.3.1 (Sample Preparation and Grouping): What was the protocol based on?
Item 2.3.2 (PMA Treatment): This item is not well written and must be revised.
Item 2.3.4 (Preparation of 16S rDNA Library): It has to explain the choice of methods and primers used.
Results
It would be interesting for the authors to address how pig and duck skin is treated in China. Did the study replicate realistic scenarios regarding the processing of these foods? Why weren't the analyses conducted immediately after the applied treatment? Could the one-week incubation period have an influence on the predominance of certain genera?
There was a lack of discussion regarding the reason why certain genera prevailed after the thermal treatment. Is it due to thermophilic characteristics? What is the significance of evaluating the genera that prevail at different temperatures when cooking needs to reach a minimum required temperature to be considered safe?
Regarding the previous revision made by the authors (highlighted in yellow), it requires careful review since there are several grammatical errors. Additionally, I believe the text is overly lengthy and could benefit from simplification. Moreover, it's important to avoid using the term "conclusion" within the discussion section, as there is a specific section dedicated to the conclusion of the study.
Caution is required when stating that pathogenic bacteria were addressed, as the identification was only conducted at the genus level, and a significant number of those identified were opportunistic pathogens.
Conclusion (not Concliusions)
The conclusion should not be divided into sections reiterating the results, but rather presented as a single paragraph effectively summarizing the findings of the study.
Comments on the Quality of English Language
Moderate editing of English language required.
Author Response
Please see the attachment

Reviewer 2 Report (New Reviewer)
Comments and Suggestions for Authors
This manuscript analyzes microbial communities from pork and duck skins, two products used in the Chinese kitchen. The research question of this manuscript is straightforward, however, their description of materials and methods could be better, and their contribution in terms of results and discussion section is scarce, thus needing to be improved before its publication in Microorganisms journal.
I've included my detailed comments and suggestions in the attached file, so please revise them carefully.

Extensive editing of English language required
Round 2
Reviewer 2 Report (New Reviewer)
Comments and Suggestions for Authors
The authors have addressed all comments and suggestions well. However, the manuscript needs a cautious review. For example, the labels of figures 2-5 are so small. Please improve them.
Also, the manuscript has some typos; please revise them all i.e. Concliusion
Comments on the Quality of English LanguageMinor editing of English language required
Author Response
Thank you for your advice. We corrected the error in the article. We hope you will look at it further. The following documents are the latest revisions

This manuscript is a resubmission of an earlier submission. The following is a list of the peer review reports and author responses from that submission.
Round 1
Reviewer 1 Report
Comments and Suggestions for Authors
I suggest the corrections or adding details as presented in the following:
Line 36 page 1 - ".....benefits to the human body [2-3]. Pork skin...."
Lines 104-107 - I do not understand where was obtain "The One-Step DNA Extraction Kit", from Tianenz Gene Technology Co., Ltd., Tianjin or Beijing?
Line 110 - Please explain the acronym "PMA". I assume that PMA means - Phorbol 12-myristate 13-acetate, but please confirm!
Line 136 - Please detail "sterile saline solution" (chemical salt and concentration). These details are important for bacteria to multiply and proliferate - and exactly this is the subject of the paper study!
Line 147 - Please mention the temperature of incubation of bacterial solution!
Line 140 - "sterile saline solution" - please detail by mentioning the chemical salt and concentration.
Line 155 - - "sterilized saline solution" - please detail by mentioning the chemical salt and concentration.
Line 290 - ", while in pork skin......." unfinished sentence!
Line 313 - "Overall, pork skin..."
Comments on the Quality of English Language
Mild English refresh could help (a second reading for a more direct construction for some sentenze in the text)
Reviewer 2 Report
Comments and Suggestions for Authors
The research content is novel and thought to be of interest to the reader. However, the results of data analysis are simple and the discussion part is insufficient.
Line 175, Is figure 1 correct?
Line 205, Recent studies favor ASV over OTU. Why did the authors use OTU?
Line 211, Why did the authors use Greengenes as 16S rRNA database?
Write genus in italics.
All figures are in low resolution so it is unrecognizable.
Uniform the use of space and underscore in group names.
Arrange the order of groups of figure for easy viewing
What are the alpha diversity results?
Why didn't the authors conduct a beta diversity analysis?
Additional analyzes, including statistical analysis of the gut microbiota, should be performed.